# *DRG2* Depletion Promotes Endothelial Cell Senescence and Vascular Endothelial Dysfunction

**DOI:** 10.3390/ijms23052877

**Published:** 2022-03-06

**Authors:** Anh-Nhung Le, Seong-Soon Park, Minh-Xuan Le, Unn Hwa Lee, Byung Kyun Ko, Hye Ryeong Lim, Ri Yu, Seong Hee Choi, Byung Ju Lee, Soo-Youn Ham, Chang Man Ha, Jeong Woo Park

**Affiliations:** 1Department of Biological Sciences, University of Ulsan, Ulsan 44610, Korea; leanhnhung1996@gmail.com (A.-N.L.); tptp3123@naver.com (S.-S.P.); lmxuan309@gmail.com (M.-X.L.); unnhwa@naver.com (U.H.L.); bjlee@ulsan.ac.kr (B.J.L.); 2Department of Surgery Ulsan University Hospital, University of Ulsan, Ulsan 44033, Korea; byungkyuko@naver.com; 3Research Strategy Office and Global Relation Center of Korea Brain Research Institute, Daegu 41062, Korea; hrsz@kbri.re.kr; 4Neurovascular Biology Laboratory, Neurovascular Unit Research Group, Korea Brain Research Institute, Daegu 41062, Korea; haru_y@kbri.re.kr; 5RopheLBio, B102, Seoul Forest M Tower, 31, Seongdong-Gu, Seoul 04778, Korea; nacchal2@naver.com; 6Department of Radiology, Sungkyunkwan University Kangbuk Samsung Hospital, Seoul 03181, Korea

**Keywords:** *DRG2*, endothelial cells, senescence, angiogenesis, vascular dysfunction

## Abstract

Endothelial cell senescence is involved in endothelial dysfunction and vascular diseases. However, the detailed mechanisms of endothelial senescence are not fully understood. Here, we demonstrated that deficiency of developmentally regulated GTP-binding protein 2 (*DRG2*) induces senescence and dysfunction of endothelial cells. *DRG2* knockout (KO) mice displayed reduced cerebral blood flow in the brain and lung blood vessel density. We also determined, by Matrigel plug assay, aorta ring assay, and in vitro tubule formation of primary lung endothelial cells, that deficiency in *DRG2* reduced the angiogenic capability of endothelial cells. Endothelial cells from *DRG2* KO mice showed a senescence phenotype with decreased cell growth and enhanced levels of p21 and phosphorylated p53, γH2AX, senescence-associated β-galactosidase (SA-β-gal) activity, and senescence-associated secretory phenotype (SASP) cytokines. *DRG2* deficiency in endothelial cells upregulated arginase 2 (*Arg2*) and generation of reactive oxygen species. Induction of SA-β-gal activity was prevented by the antioxidant N-acetyl cysteine in endothelial cells from *DRG2* KO mice. In conclusion, our results suggest that *DRG2* is a key regulator of endothelial senescence, and its downregulation is probably involved in vascular dysfunction and diseases.

## 1. Introduction

Endothelial cells lining the interior surface of blood vessels regulate blood flow in vascular circulation [1], playing an important role in vascular function [2]. Angiogenesis, the formation of new capillaries from pre-existing blood, is primarily undertaken by endothelial cells [3]. Endothelial dysfunction impairs angiogenesis and contributes to the increased prevalence of cardiovascular diseases [4]. Therefore, understanding the basis of endothelial dysfunction has important implications for understanding and managing cardiovascular disease.

Cellular senescence is characterized by cell-cycle arrest and pro-inflammatory changes in gene expression [5]. The senescence of endothelial cells leads to endothelial dysfunction [6]. Senescence can be induced by various kinds of stimuli, including ionizing radiation [7], telomere dysfunction [8] or reactive oxygen species (ROS) [9]. Among these, oxidative stress plays a central role in the development of cellular senescence [10]. Oxidative stress occurs when the production of ROS overwhelms endogenous antioxidant systems and/or when endogenous antioxidant systems are impaired [11,12]. Although oxidative stress is a major factor in the onset of senescence [13], the specific mechanisms underlying ROS-induced endothelial senescence are not clear.

In the endothelium, endothelial nitric oxide synthase (eNOS) uses l-arginine as a substrate to produce nitric oxide (NO), which plays a protective physiological role in the vasculature [14]. Arginase 2 (*Arg2*), a predominant isoform of arginase in endothelial cells [15], competes directly with eNOS for l-arginine to inhibit NO synthesis [16]. Upregulation of *Arg2* decreases intracellular l-arginine content to reduce NO production and increases the production of ROS by eNOS uncoupling [17,18]. Thus, *Arg2* plays an important role in endothelial dysfunction with implications in vascular disease [19].

Developmentally regulated GTP-binding proteins (DRGs) constitute a subfamily of the GTPase superfamily [20]. The DRG subfamily consists of two closely related proteins, DRG1 and *DRG2* [21]. DRG1 and *DRG2* interact with different molecules, DRG family regulatory protein 1 (DFRP1) and DFRP2, respectively [22], suggesting that they have distinct functions. Previously, we found that overexpression of *DRG2* affects cell proliferation and apoptosis in Jurkat human T cells [23,24], inhibits T_H_17 differentiation, and ameliorates experimental autoimmune encephalomyelitis in mice [25]. We also learned that *DRG2* interacts with Rab5 on endosomes and is required for Rab5 inactivation on endosomes and for recycling of transferrin (Tfn) to the plasma membrane [26]. Recently, it was shown that *DRG2* knockdown induces Golgi fragmentation [27] and mitochondrial dysfunction [28], decreases the stability of Rac1-positive membrane tubules in cancer cells [29], and suppresses VEGF-A production in melanoma cells, leading to inhibition of tumor angiogenesis [30]. Knocking out *DRG2* impairs dopamine release from dopamine neurons in the mouse brain [31]. Together, these data demonstrate that *DRG2* is an important regulator of signal pathways for cell growth, differentiation, and/or vesicle trafficking. However, little is known about the functional role of DRGs in endothelial cells.

In this study, we examined the role of *DRG2* in the regulation of endothelial-generated ROS and endothelial senescence. We demonstrated that *DRG2* depletion increases NADPH oxidase 2 (*NOX2*) expression, ROS generation, and senescence in endothelial cells. In addition, we found that *DRG2* deficiency reduces the angiogenic activity of endothelial cells, lung blood vessel density, and blood flow in the cerebral brain. Together, these results suggest that *DRG2* deficiency leads to endothelial dysfunction and impaired angiogenesis through upregulation of ROS and senescence in endothelial cells.

## 2. Results

### 2.1. DRG2^−/−^ Mice Exhibit Decreased Microvascular Circulation and Blood Vessel Density

We previously demonstrated that *DRG2* depletion in cancer cells suppresses tumor angiogenesis [30]. In the current study, we further investigated the effects of *DRG2* depletion on endothelial cell behavior in mice. As a functional parameter, we determined whether *DRG2* deficiency affects the microvascular circulation by evaluating the velocity of blood flow in the cerebral cortex of wild-type and *DRG2*^−/−^ mice under normal conditions using laser speckle flowmetry. The results showed that the cerebral blood flow (CBF) of the ipsilateral cortex of *DRG2*^−/−^ mice decreased to 72% of that in wild-type mice (Figure 1A). The time course of the relative changes in CBF for vascular compartments is shown in Figure 1B. No significant change in CBF level of wild-type or *DRG2*^−/−^ mice were detected over the periods observed (Figure 1B), which suggests that microvascular circulation was stable in both wild-type and *DRG2*^−/−^ mice. The CBF level of *DRG2*^−/−^ mice was significantly lower than that of wild-type mice over the periods observed (20 s) (Figure 1B). We next determined whether *DRG2* deficiency affects the blood vessel density in lung tissues. The sections of lung tissues from wild-type and *DRG2*^−/−^ mice were stained to detect CD31, a vascular endothelial cell marker. A representative lung section stained with anti-CD31 antibody is shown in Figure 1C,D, with quantitation of vessel area and microvessel number. The immunohistological analysis confirmed significantly reduced blood vessel density, measured as a reduced percentage in CD31-positive vessel area (Figure 1C), as well as a reduction in microvessel number (Figure 1D) in the lung of *DRG2*^−/−^ mice compared with that of wild-type mice. These results indicate that *DRG2* deficiency decreases microvascular circulation and blood vessel density in mice. Defects in microvascular circulation and blood vessel density can lead to an increase in cardiovascular diseases [4]. Consistently, we reported growth retardation and shortened lifespan in *DRG2*^−/−^ mice [31]. We next determined whether *DRG2* deficiency affects the reproduction of mice. We found that the number of progeny per litter of *DRG2*^−/−^ mice was significantly less than that of WT mice (Figure 1E).

### 2.2. DRG2 Deficiency Inhibits Angiogenic Functions of Endothelial Cells 

The endothelial cell lining of blood vessels is a critical component of many physiological functions [32]. To determine the effect of *DRG2* deficiency on the angiogenic function of endothelial cells, three studies were conducted. First, we investigated the influence of *DRG2* deficiency on the formation of capillary-like tubes in primary mouse lung endothelial cells (mLECs) after the addition of recombinant VEGF-A. VEGF-A was found to stimulate tube formation in both wild-type and *DRG2*^−/−^ mLECs (Figure 2A,B). However, the total tube lengths in *DRG2*^−/−^ mLECs treated with 100 ng/mL VEGF-A were significantly decreased compared with those of the wild-type (Figure 2A,B), demonstrating that *DRG2* deficiency impaired endothelial cell tube formation in vitro. 

Second, we assessed the effect of *DRG2* deficiency on the angiogenic potential of mice, by in vivo Matrigel plug assay. Matrigel was subcutaneously injected into wild-type and *DRG2*^−/−^ mice, and the implants were removed after 10 days to evaluate neovascularization. To allow efficient vascularization, angiogenic growth factor VEGF-A was suspended in the Matrigel before injection. Controls did not contain the VEGF-A supplement. A red-colored plug filled with red blood cells (RBCs) indicated the formation of new blood vessels inside the Matrigel. As shown in Figure 2C, the red color of plugs from *DRG2*^−/−^ mice was much lighter than that from wild-type mice, suggesting the formation of fewer blood vessels. Matrigel plugs were harvested, sectioned, and analyzed for capillary by immunohistochemical (IHC) staining with anti-CD31 antibody. The overall vascular densities in the Matrigel plugs are shown in Figure 2D. *DRG2*^−/−^ mice exhibited lower neovascularization of the Matrigel implants upon angiogenic challenge with VEGF compared with wild-type mice (Figure 2D). Without the addition of VEGF-A (i.e., controls), the implants remained largely avascular. In addition, whereas the vascular endothelial structures detected in the plugs from wild-type mice were orderly, the blood vessels in the plugs from *DRG2*^−/−^ mice were poorly organized (Figure 2D). 

Third, we prepared aortas from wild-type and *DRG2*^−/−^ mice and determined the effect of *DRG2* deficiency on endothelial cell outgrowth by performing aortic ring assays. The aortic ring assays analyzed the newly formed endothelial cell network protruding from aortic explants. A comparison of the assay can be found in Figure 2E. Endothelial cell tubes protruding from aortic explants of *DRG2*^−/−^ mice formed a less complex endothelial network compared with the rather dense network emerging from explants of wild-type mice. In addition, the overall length of the endothelial cell sprouts emerging from the aortic explants of *DRG2*^−/−^ mice was significantly lower than that of wild-type mice (Figure 2F). Collectively, these results suggest that *DRG2* deficiency leads to defective angiogenesis. 

### 2.3. DRG2 Deficiency Decreases Proliferation and Enhances Senescence of Endothelial Cells

Cell proliferation is essential for endothelial cells to adequately perform their angiogenic functions; therefore, the senescence of endothelial cells impairs angiogenesis [33]. The defects in the angiogenic function of *DRG2*^−/−^ endothelial cells prompted us to investigate whether *DRG2* deficiency affects the proliferation of endothelial cells. We first determined the effect of *DRG2* deficiency on mLECs proliferation using an MTS assay. The *DRG2*^−/−^ mLECs demonstrated decreased proliferation compared with that of *DRG2*^+/+^ mLECs (Figure 3A). A Ki67 staining study further supported the decreased proliferation in *DRG2*^−/−^ mLECs (Figure 3B). Since apoptosis can contribute to the numbers of endothelial cells, we examined apoptotic activity in mLECs by Annexin-V staining. However, there was no significant difference in apoptosis between wild-type and *DRG2*^−/−^ mLECs (Figure 3C). We next asked whether *DRG2* deficiency affects the senescence of endothelial cells. Senescent cells have increased activity of senescence-associated β-gal (SA-β-gal) [34], a well-defined biomarker for cellular senescence. We evaluated SA-β-gal activity in wild-type and *DRG2*^−/−^ mLECs. The number of SA-β-gal-stained cells was significantly increased in *DRG2*^−/−^ mLECs compared with that of *DRG2*^+/+^ mLECs (Figure 3D). One of the defining features of senescent cells is cell cycle arrest in G1 [7]. We examined whether *DRG2* deficiency induces cell cycle arrest in mLECs. Cell cycle distribution analysis revealed an increased accumulation of G1 phase cells in *DRG2*^−/−^ mLECs (Figure 3E). To determine whether *DRG2*^−/−^ mice show senescence phenotype, we collected uteruses from wild-type and *DRG2*^−/−^ mice and stained them for SA-β-gal activity. We could detect stronger SA-β-gal activity in *DRG2*^−/−^ uteruses than wild-type uteruses (Figure 3F). These results indicate that *DRG2* deficiency decreases proliferation and increases the senescence of endothelial cells. 

### 2.4. DRG2 Deficiency Enhances DNA Damage and Senescence Induced by Oxidative Stress

Oxidative stress causes DNA damage [35] and cellular senescence [13,36]. Since H_2_O_2_ is a major ROS generated during oxidative stress [37,38], we examined whether *DRG2* deficiency affects the senescence response to oxidative stress using H_2_O_2_. H_2_O_2_ significantly increased the number of SA-β-gal-stained cells in both wild-type and *DRG2*^−/−^ mLECs (Figure 4A). However, after H_2_O_2_ treatment, the number of SA-β-gal-stained cells in *DRG2*^−/−^ mLECs was much higher than that of wild-type mLECs (Figure 4A). Accumulated DNA damage activates p53, leading to cellular senescence [39]. An increase in phosphorylated histone H2AX (γH2AX) was considered as a biomarker of cellular senescence [40]. We examined the effect of *DRG2* deficiency on oxidative stress-induced DNA damage by analyzing the levels of phosphorylated p53 and γH2AX. After H_2_O_2_ treatment, *DRG2*^−/−^ mLECs showed a significantly higher level of total and phosphorylated p53 than wild-type mLECs did (Figure 4B). Consistently, the level of γH2AX induced by H_2_O_2_ was higher in *DRG2*^−/−^ mLECs than in WT mLECs (Figure 4C). Even in the absence of H_2_O_2_ treatment, *DRG2*^−/−^ mLECs showed higher levels of phosphorylated p53 and γH2AX than WT mLECs did (Figure 4B,C). We also treated mLECs with DNA damaging agents doxorubicin and etoposide and analyzed the expression level of phosphorylated p53. Consistent with H_2_O_2_, *DRG2*^−/−^ mLECs showed a significantly higher level of phosphorylated p53 than wild-type mLECs after treatment with doxorubicin (Figure 4D) and etoposide (Figure 4E). In addition, doxorubicin and etoposide induced a higher level of p21 in *DRG2*^−/−^ mLECs than wild-type mLECs (Figure 4D,E). These results suggest that *DRG2* deficiency increases the levels of p21, γH2AX, and phosphorylated p53 induced by oxidative stress in mLECs, which means that *DRG2* may play a role in suppressing their levels in mLECs. Senescent cells are characterized by production of inflammatory cytokines, immune modulators, growth factors, and proteases, which comprise the senescence-associated secretory phenotype (SASP) [41]. To determine if *DRG2* deficiency would also induce components of the SASP, levels of *IL-6*, *CXCL10*, and *IFN-β1* in wild-type and *DRG2*^−/−^ mLECs were measured after etoposide treatment using qPCR. In the absence of etoposide treatment, both wild-type and *DRG2*^−/−^ mLECs expressed very low levels of these cytokines and *DRG2*^−/−^ mLECs expressed higher level of only IL-6 than wild-type mLECs did. After etoposide treatment, expression levels of three cytokines were enhanced in both wild-type and *DRG2*^−/−^ mLECs and *DRG2*^−/−^ mLECs expressed significantly higher levels of these cytokines than wild-type mLECs did (Figure 4F). These data suggest that DNA damage and cellular senescence are enhanced by oxidative stress in *DRG2*^−/−^ mLECs.

### 2.5. DRG2 Deficiency Affect the Expression of Antioxidant Genes

Previously, we reported that *DRG2* deficiency induces mitochondrial dysfunction and increases intracellular ROS in cancer cells [28]. Consistently, *DRG2*^−/−^ mLECs showed a decrease in mitochondrial membrane potential compared with wild-type mLECs (Figure 5A). We also confirmed an increase in intracellular ROS level before and after doxorubicin treatment in *DRG2*^−/−^ mLECs compared to wild-type mLECs (Figure 5B). Since oxidative stress is one of the major factors causing the onset of senescence [13], we used ROS scavenger N-acetylcysteine (NAC) to test whether increased levels of ROS lead to increased senescence in *DRG2*^−/−^ mLECs. NAC treatment effectively reduced the number of SA-β-gal-stained cells in both wild-type and *DRG2*^−/−^ mLECs (Figure 5C), indicating that increased ROS is one of the major causative factors for enhanced senescence in *DRG2*^−/−^ mLECs. To combat oxidative stress, cells produce several antioxidant enzymes [42]. We next examined whether *DRG2* deficiency affects the expression of the antioxidant genes, superoxide dismutase 1 (*SOD1*) and *SOD2*. The expression levels of *SOD1* (Figure 5D) and *SOD2* (Figure 5E) were significantly lower in *DRG2*^−/−^ mLECs both before and after H_2_O_2_ treatment than in wild-type mLECs, implicating the reduced expression of antioxidant genes to play a role in increased ROS level and senescence in *DRG2*^−/−^ mLECs. 

### 2.6. DRG2 Deficiency Decreases eNOS Expression and NO Production in Endothelial Cells

Since eNOS-derived NO functions are an anti-senescent factor in ECs [43], we examined the effect of *DRG2* deficiency on the eNOS/NO pathway. There was no significant difference in the expression of *eNOS* between *DRG2*^−/−^ mLECs and wild-type mLECs (Figure 5F). However, NO production was significantly decreased in *DRG2*^−/−^ mLECs compared to wild-type mLECs (Figure 5G). eNOS produces NO by catalyzing L-arginine to L-citrulline. However, when the substrate L-arginine is depleted by the enzyme arginase (Arg), eNOS produces ROS instead of NO, which was referred to as eNOS coupling. The increased ROS generation leads to endothelial dysfunction [44,45]. We next examined whether *DRG2* deficiency affects the expression of *Arg2* in endothelial cells. The expression level of *Arg2* was significantly higher in *DRG2*^−/−^ mLECs both before and after etoposide treatment than in wild-type mLECs (Figure 5H). Collectively, these results suggest that the induction of senescence by *DRG2* deficiency in mLECs is associated with modulated expression of *Arg2*, which affects the intracellular levels of NO and ROS.

## 3. Discussion

Our previous data suggested that *DRG2* in cancer cells is involved in tumor angiogenesis [30]. However, the role of *DRG2* in endothelial cells remains to be explored. In the present study, we investigated the role of *DRG2* in endothelial cell functions, especially angiogenic functions, using *DRG2*^−/−^ mice. Here, we provide evidence that *DRG2* is involved in angiogenesis and vascular remodeling, in that (a) *DRG2*^−/−^ mice showed decreased blood flow in the brain. Since regulation of CBF is controlled by cerebral endothelium [46], reduction in the CBF in *DRG2*^−/−^ mice implicates the role of *DRG2* in endothelium function; (b) *DRG2* deficiency reduced pulmonary blood vessel density. The intensity of CD31 staining in the lung of *DRG2*^−/−^ mice was significantly reduced compared to that of wild-type mice; (c) *DRG2* deficiency inhibited neovascularization of subcutaneous Matrigel implants. When the Matrigel plugs containing the angiogenic growth factor VEGF were transplanted subcutaneously, *DRG2*^−/−^ mice exhibited low and poorly organized neovascularization of the Matrigel implant compared to wild-type mice. These results suggest that *DRG2* deficiency leads to a defect in angiogenesis; (d) *DRG2* deficiency impaired microvessel outgrowth in an ex vivo aorta ring assay. In aortic ring assays, the branching of newly formed sprouts was significantly reduced in *DRG2*^−/−^ aorta explants compared to WT aorta explants; and (e) *DRG2* deficiency reduced the in vitro tube-forming capability of endothelial cells. When treated with VEGF, lung endothelial cells from *DRG2*^−/−^ mice showed a decrease in the formation of tube-like structures compared to those from wild-type mice. Collectively, our results suggest that *DRG2* deficiency induces dysfunction of endothelial cells, especially in angiogenesis.

Cell proliferation is essential for endothelial cells to adequately perform their angiogenic functions and senescence of endothelial cells can contribute to endothelial dysfunction and impair angiogenesis [33,47]. We reported previously that *DRG2* deficiency increases intracellular ROS levels in cancer cells [28]. Consistently, in this experiment, we also found that *DRG2*^−/−^ endothelial cells produce enhanced levels of ROS compared to wild-type endothelial cells. ROS can induce DNA damage [35] and cellular senescence [13,36]. In the present work, we found that *DRG2* deficiency enhanced SA-β-gal activity, a biomarker for senescence [34], in endothelial cells. The increase in SA-β-gal activity in *DRG2*^−/−^ endothelial cells was attenuated by concomitant application of NAC, a ROS scavenger, indicating that increased levels of ROS led to increased senescence in *DRG2*^−/−^ endothelial cells. A characteristic feature of cellular senescence is cell cycle arrest in the G1 phase [7] and the SASP [41] involving increased expression of inflammatory molecules [48]. In the present work, we also found that *DRG2*^−/−^ endothelial cells expressed higher levels of inflammatory molecules such as *IL-6*, *CXCL-10*, and *IFN-β1* than did wild-type endothelial cells, providing further evidence supporting the induction of senescence in *DRG2*^−/−^ endothelial cells. Excessive ROS production can induce DNA damage [36,49,50] and activate the p53-p21^WAF1/CIP1^ senescence pathway [51,52,53]. Consistently, we observed cell cycle arrest in the G1 phase (Di Leonardo et al., 1994) and increased levels of DNA damage marker γH2AX [40], p21 and phosphorylated p53 in *DRG2*^−/−^ endothelial cells compared to wild-type endothelial cells. Collectively, our results indicate that *DRG2* deficiency increases intracellular ROS, which leads to DNA damage and senescence in endothelial cells. The proliferation of endothelial cells is one of the key steps of the VEGF-mediated angiogenic process [54]. *DRG2* deficiency-induced senescence may limit the capacity of VEGF to promote angiogenesis in *DRG2*^−/−^ endothelial cells.

How does *DRG2* deficiency increase ROS in endothelial cells? eNOS plays a key role in increasing vascular ROS production [55]. It is the predominant NOS form in the vasculature and normally oxidizes its substrate L-arginine to produce the vasoprotectant molecule NO [56]. However, under limited-L-arginine, eNOS produces ROS instead of NO, referred to as eNOS uncoupling [57,58]. Endothelial cells express arginases that can compete with eNOS for L-arginine and, if highly expressed, “starve” eNOS. Arginase exists in two isoforms; in human endothelial cells, *Arg2* seems to be the predominant isozyme [15,59]. Upregulation of *Arg2* was found to decrease NO generation and increase ROS production via eNOS uncoupling [45]. In the present work, we found that, even though *DRG2* deficiency did not affect *eNOS* expression, it downregulated the intracellular NO level in endothelial cells. When we analyzed the expression level of *Arg2*, we found that *DRG2*^−/−^ endothelial cells produced a higher level of *Arg2* than did wild-type endothelial cells, suggesting that *DRG2*^−/−^ endothelial cells produce ROS via eNOS uncoupling. Although our study did not determine other sources of ROS and the effect of *Arg2* inhibition on ROS production, our results suggest that eNOS uncoupling is an important contributor to ROS production and senescence induction in *DRG2*^−/−^ endothelial cells.

In summary, we demonstrated that *DRG2* deficiency can induce endothelial dysfunction, such as reduced blood flow and a decrease in angiogenesis in vivo, ex vivo, and in vitro. *DRG2* deficiency increased ROS production in endothelial cells through eNOS uncoupling and induced endothelial senescence. The enhanced endothelial senescence can contribute to the dysfunction found in the *DRG2*-deficient endothelium. Even though we did not determine whether there was an enhanced frequency of vascular diseases in *DRG2*^−/−^ mice, *DRG2* deficiency can increase ROS and, thus, endothelial dysfunction and lead to vascular diseases. Further studies are required to explore in more detail the roles of *DRG2* in DNA damage, endothelial senescence, and vascular diseases.

## 4. Materials and Methods

### 4.1. Cell Culture

Mouse lung endothelial cells (mLECs) were isolated by methods as previously described [30]. Briefly, cell suspensions were prepared by digesting mouse lungs in collagenase and incubated with anti-CD31 monoclonal antibody for 30 min at 4 °C. Cells were pulled down using magnetic beads coated with sheep anti-IgG. After washing four times, cells were digested with trypsin-EDTA to detach the beads. Cells were cultured in Dulbecco’s Modified Eagle Medium (DMEM) (Thermo Fisher Scientific, Waltham, MA, USA) supplemented with 15% fetal bovine serum (FBS; Thermo Fisher Scientific, Waltham, MA, USA), heparin (100 μg/mL) (Sigma, St. Louis, MO, USA), 100 U/mL penicillin, and 100 μg/mL streptomycin (Thermo Fisher Scientific) in a 2% gelatin (Sigma)-coated plate at 37 °C and 5% CO_2_. 

### 4.2. Mice and Animal Research Ethics

*DRG2*^−/−^ mice were as described previously [26]. Mice were bred in the animal facility at the University of Ulsan, and offspring were born and housed in the same room under specific pathogen-free conditions. All animal procedures were approved by the Institutional Animal Care and Use Committee of the Meta-inflammation Research Center (permit number JWP-21-020). All surgery was performed under sodium pentobarbital anesthesia, and all efforts were made to minimize suffering.

### 4.3. Cortical Cerebral Blood Flow Measurement 

Real-time two-dimensional cerebral blood flow (CBF) was monitored using a laser speckle contrast imager (PeriCam PSI HR System, Perimed, Sweden). Mice were anesthetized and placed in the prone position and the skull was exposed through a cut in the skin at the parietal midline. A scanning camera was placed above the head at a working distance of 10 cm from the skull surface and illuminated with a laser diode (785 nm) to penetrate the brain. To evaluate CBF changes, the region-of-interest (ROI) included the cortical area supplied by the middle cerebral artery.

### 4.4. In Vivo Matrigel Plug Assays

This assay was performed by methods described previously with modification [60]. Mice were injected subcutaneously with 0.5 mL Matrigel (BD Biosciences, San Jose, Santa Clara, CA, USA) containing 100 ng recombinant murine VEGF (R&D Systems, Minneapolis, MN, USA). Controls did not contain VEGF. After 10 days, the mice were sacrificed and plugs were harvested from underneath the skin. The plugs were fixed, embedded, and sectioned. To visualize capillaries, samples were immunohistochemically stained with anti-CD31 antibody (sc-376764, Santa Cruz, Santa Cruz, CA, USA). 

### 4.5. Aortic Ring Assay

Aortas were prepared from wild-type and *DRG2* KO mice, and aortic ring assays were performed as described previously with modification [61]. Briefly, 0.5 mm mouse aortic rings were embedded in 3-dimensional growth factor reduced Matrigel (354230, Corning, Corning, NY, USA). Opti-MEM (31985062, Thermo Fisher Scientific) containing 30 ng/mL VEGF (450-32, PeproTech, Cranbury, NJ, USA) was added to induce vessel outgrowth. Sprouting vessels were counted at day 7 after incubation and photographed using a Nikon Eclipse TS 100 (Nikon, Tokyo, Japan) phase-contrast microscope. 

### 4.6. Immunohistochemistry and Quantification of Lung Angiogenesis 

Immunohistochemical analysis was performed on 4-μm-thick tissue sections cut from formalin-fixed paraffin-embedded surgical specimens. The sections were stained with hematoxylin and eosin (HE) or with anti-CD31 antibody (1:200 dilution; DAKO, Glostrup, Denmark) using a VENTANA BenchMark XT automated staining device (Ventana Medical System, Tucson, AZ, USA), according to the manufacturer’s instructions. The Vessel area was determined by CD31 positive staining normalized to the total area fraction. The number of microvessels was counted as described previously [62]. Briefly, the most vascularized areas were selected and counted in a ×400 field. Any single or cluster of endothelial cells that were separated clearly from adjacent microvessels was considered as one countable microvessel. The average counts from the three most vascularized areas were recorded in each case to measure the degree of angiogenesis. 

### 4.7. Endothelial Tube Formation In Vitro Assays

The in vitro angiogenic activity of mLECs derived from wild-type and *DRG2* KO mice was determined by Matrigel tube formation assay. Matrigel (354230, Corning) was added to each well (50 μL per well) of a 96-well plate and allowed to solidify at 37 °C for 30 min. Then, 1 × 10^4^ mLEC cells in 100 μL of DMEM were placed in each well and incubated at 37 °C for 6 h with or without addition of 100 ng/mL of mouse recombinant VEGF (BD Biosciences). The ability of the cells to form endothelial tubes was evaluated under phase-contrast microscopy using a Nikon Eclipse TS100 inverted microscope equipped with a Nikon DXM-1200 digital camera (Nikon, Tokyo, Japan). Images of tube morphology were obtained in five random microscopic fields per sample at ×40 magnification, and the cumulative tube lengths were measured by Image-Pro Plus software (Media Cybernetics, Rockville, MD, USA). 

### 4.8. Colony-Forming Assay

Clonogenic assays were carried out as described previously [63]. Briefly, 1600 cells were seeded into a 60 mm plate and incubated for 2 h at 37 °C. They were then incubated for a further 2 weeks with or without addition of 8 μM or 16 μM doxorubicin (Sigma). Cells were fixed with 3.7% formalin for 10 min and stained with 0.1% crystal violet for 20 min. After washing in water, the colonies were counted. 

### 4.9. Cell Cycle Analysis

Cells were harvested with trypsin-EDTA (Gibco) and fixed for 30 min in ice-cold 70% ethanol. After washing twice with PBS, cells were incubated for 30 min at 37 °C with 100 µg/mL RNase (R5125, Sigma). A final concentration of 50 µg/mL propidium iodide (PI) (P4170, Sigma) was added and DNA was measured with a flow cytometer (BD FACSCanto II, BD Biosciences).

### 4.10. Ki-67 Analysis

Cells were fixed in 4% paraformaldehyde for 10 min and permeabilized with 0.5% Triton X-100 for 10 min. They were then incubated with Alexa Fluor 488-conjugated anti-Ki67 antibody (#11882, Cell Signaling, Danvers, MA, USA) for 1 h. Confocal images were obtained using an Olympus 1000/1200 laser-scanning confocal system (Olympus, Shinjuku, Japan). All images were saved as TIFF files, and their contrast was adjusted with Image J (NIH, Bethesda, MD, USA, v1.19m). Quantification was performed by counting the number of Ki-67-positive cells.

### 4.11. MTS Assay

MTS assay was performed using the CellTiter 96 Aqueous One Solution Cell Proliferation Assay Kit (3580, Promega, Madison, WI, USA). Briefly, cells were plated in triplicate at 1 × 10^4^ mLEC cells/well in 96-well culture plates in DMEM. At 24 h after plating, cell viability was measured using the MTS assay, according to the protocol supplied by the manufacturer.

### 4.12. Annexin-V Staining

Apoptotic cells were stained using an Annexin-V-FLUOS staining kit (#6592, Cell Signaling), according to the protocol supplied by the manufacturer. Annexin-V-stained cells were analyzed for fluorescence intensity using a FACSCanto II Flow Cytometer (BD Biosciences).

### 4.13. Senescence-Associated β-Galactosidase (SA β-gal) Activity

SAβ-gal activity was measured using a Senescence Cells Histochemical Kit (CS0030, Sigma) according to the protocol supplied by the manufacturer. Briefly, cells were incubated for 24 h with or without addition of 200, 400, and 800 μM H_2_O_2_ (H1009-100ML, Sigma) or 5 mM N-acetyl-cysteine (NAC) (A9165-25G, Sigma). After fixation with Fixation buffer (2% formaldehyde and 0.2% glutaraldehyde) for 6 min, they were incubated with Staining Mixture at 37 °C overnight. Cells were examined under a ZEISS Primovert microscope (ZEISS, Germany) with ZEISS Axiocam ERc 5s camera (ZEISS, Germany). The cytosol in senescent cells was stained blue. To determine the SA-β-gal activity in uteruses, uteruses were collected from mice, fixed with 10% formaldehyde (*v*/*v*) (F1635, Sigma) for 10 min at room temperature, and incubated with Staining mixture for 24 h at 37 °C.

### 4.14. Confocal Microscopy 

Cells were seeded on 35-mm-diameter confocal dishes (200350, SPL). For detection of mitochondrial membrane potential, cells were labeled with 250 nM tetramethylrhodamine, methyl ester, perchlorate (TMRM, T668, Molecular Probes) in DMEM for 30 min. Nuclei were stained with 4′,6-diamidino-2-phenylindole (DAPI) (Sigma-Aldrich, St. Louis, MO, USA). Confocal images were obtained using an Olympus 1000/1200 laser-scanning confocal system (Olympus, Shinjuku, Japan) equipped with a 100× Plan Apochromat NA/1.4 oil objective and the appropriate filter combination. All images were saved as TIFF files, and their contrast was adjusted with Image J (NIH, Bethesda, MD, USA, v1.19m). 

### 4.15. SDS-PAGE Analysis and Immunoblotting 

Proteins were resolved by SDS-PAGE and transferred onto nitrocellulose membranes (10600001, GE Healthcare, Piscataway, NJ, USA). The membranes were then probed with the appropriate dilutions of anti-*DRG2* (#14743-1-AP, Proteintech), anti-β-actin (A5441, Sigma), anti-p53 (DO-1, sc-126, Santa Cruz), anti-phospho-p53 (Ser15, #9284), anti-caspase-3 (#9662), anti-PARP (46D11, #9532), anti-phospho-histone H2A.X (Ser139, #2577), anti-p38 MAPK (#9212), anti-phospho-p38 MAPK (Thr180/Tyr182, #9211), anti-p21 Waf1/Cip1 (DCS60, #2946), and anti-α-Tubulin (#2144, Cell Signaling). Immunoreactivity was detected using Amersham ECL Prime (RPN2232, Cytiva). Membranes were exposed at multiple time points to ensure that the images were not saturated. If required, the band densities were analyzed with NIH image software and normalized by comparison with the densities of internal control β-actin bands. 

### 4.16. Quantitative Real-Time PCR and Semi-qRT-PCR 

Total RNA was extracted from cells using TRIzol (Thermo Fisher Scientific). Two micrograms of total RNA were reverse transcribed using oligo-dT (79237, Qiagen, Hilden, German) and MMLV reverse transcriptase (3201, Beamsbio) according to the manufacturers’ instructions. A qRT-PCR was performed by monitoring the real-time increase in fluorescence of SYBR Green (MasterMix-R, Abm) using a StepOnePlus Real-Time PCR system (Applied Biosystems, Foster City, MI, USA). RT-PCR was performed using Taq polymerase 2× premix (Solgent, Daejeon, Korea) and the appropriate primers. PCR primer pairs were as follows: IL-6, 5′-TAGTCCTTCCTACCCCAATTTCC-3′ and 5′-TTGGTCCTTAGCCACTCCTTC-3′; VEGF-A, 5′-AGGCTGCTGTAACGATGAA-3′ and 5′-TATGTGCTGGCTTTGGTGAG-3′; Catalase, 5′-GGAGGCGGGAACCCAATAG-3′ and 5′-GTGTGCCATCTCGTCAGTGAA-3′; GPX1, 5′-GTGCAATCAGTTCGGACACCA-3′ and 5′-CACCAGGTCGGACGTACTTG-3′; SOD1, 5′-TATGGGGACAATACACAAGGCT-3′ and 5′-CGGGCCACCATGTTTCTTAGA-3′; SOD2, 5′-TGGACAAACCTGAGCCCTAAG-3′ and 5′-CCCAAAGTCACGCTTGATAGC-3′; ARG1, 5′-ACAGAACTAAGCAAACGCC-3′ and 5′-TTCATACCAGAAAGGAACTGC-3′; *ARG2*, 5′-GGCTGAAGTGGTTAGTAGAG-3′ and 5′-GGGCGTGACCGATAATG-3′; iNOS, 5′-GTTCTCAGCCCAACAATACAAGA-3′ and 5′-GTGGACGGGTCGATGTCAC-3′; eNOS, 5′-TCAGCCATCACAGTGTTCCC-3′ and 5′-ATAGCCCGCATAGCGTATCAG-3′; *DRG2*, 5′-CTCAACAGTCACACTGACAC-3′, 5′-TACCGCAACTGATAACTACA-3′; GAPDH, 5′-ACATCAAGAAGGTGGTGAAG-3′ and 5′-CTGTTG CTGTAGCCAAATTC-3′; IFN-β1: 5′-AAAGCAAGAGGAAAGATTGACG-3′ and 5′-ACCACCATCCAGGCGTA-3′; CXCL10, 5′-CCAAGTGCTGCCGTCATTTTC-3′ and 5′-GGCTCGCAGGGATGATTTCAA-3′; RPLP0, 5′-AGATTCGGGATATGCTGTTGGC-3′ and 5′-TCGGGTCCTAGACCAGTGTTC-3′.

### 4.17. Measurement of NO Production

Intracellular NO was measured using 4-amino-5-methylamino-2′,7′-difluorofluorescein (DAF-FM) diacetate (D23844, Invitrogen, Carlsbad, CA, USA). Cells were incubated in the dark with 10 μM DAF-FM diacetate for 60 min and then washed twice with media. Nuclei were stained with DAPI. Fluorescence was analyzed using a FACSCanto II Flow Cytometer (BD Biosciences) and an Olympus 1000/1200 laser-scanning confocal system (Olympus, Shinjuku, Japan).

### 4.18. Measurement of ROS Levels 

H_2_O_2_ level was measured using a ROS-Glo H_2_O_2_ Assay (G8820, Promega) according to the protocol supplied by the manufacturer. Briefly, 1 × 10^4^ cells/well were seeded in white 96 well plates and incubated for 24 h. After further incubation for 24 h with or without 8 μM or 16 μM doxorubicin (Sigma), the cells were incubated with 20 µL H_2_O_2_ substrate solution for 6 h and with 100 µL of ROS-Glo detection solution for 20 min. The luminescence signal obtained was measured using a SpectraMax L microplate reader (Molecular Devices).

### 4.19. Statistics Analysis

For statistical comparisons, *p* values were determined using Student’s *t*-test or one-way or two-way ANOVA. A *p*-value < 0.05 was considered to indicate statistical significance.

## Figures and Tables

**Figure 1 ijms-23-02877-f001:**
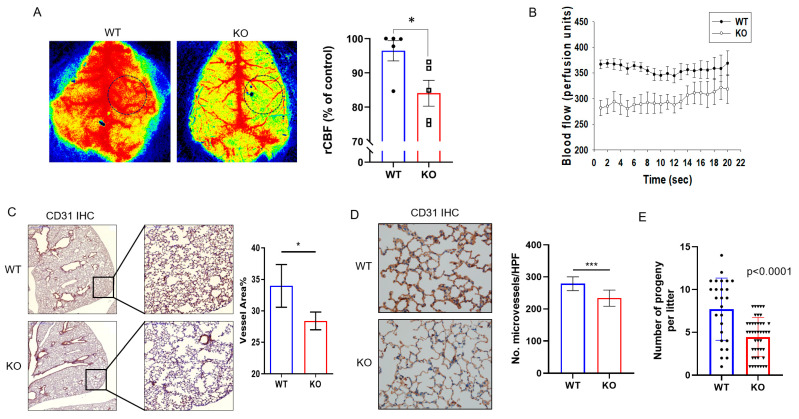
Effects of *DRG2* deficiency on cerebral blood flow (CBF) and lung blood vessels density. (**A**,**B**) Effect of *DRG2* deficiency on CBF. (**A**) Representative laser speckle images of cerebral blood perfusion in wild-type (WT) and *DRG2*^−/−^ (KO) mice. All images show areas of high and low blood perfusion as yellow–red and blue-black, respectively. CBF in the cerebral cortical area was calculated from the circular region-of-interest (ROI) (black dotted line) and quantified (n = 5 mice/group). * *p* < 0.05. (**B**) The time courses change in CBF at the circular ROI of the cerebral cortical area (n = 5 mice/group). (**C**,**D**) Effect of *DRG2* deficiency on blood vessel density in mouse lung. (**C**) Representative microscopic images of CD31 immunohistochemistry in mouse lungs (original magnification, 10× and 40×, respectively) with quantification of vessel area fraction. The data are expressed as the mean ± SEM (n = 15 areas from 2 mice in each group). * *p* < 0.05. (**D**) Representative microscopic images from hot spot areas of mouse lung immunostained for CD31 (original magnification, 400×) with quantification of microvessel number per high-power field (HPF). The data are expressed as the mean ± SEM (n = 15 areas from 2 mice in each group). *** *p* < 0.001. (**E**) Quantification of the number of progeny per litter of wild-type and *DRG2*^−/−^. The data are expressed as mean ± SD. (n = 26 for wild-type and 45 for *DRG2*^−/−^).

**Figure 2 ijms-23-02877-f002:**
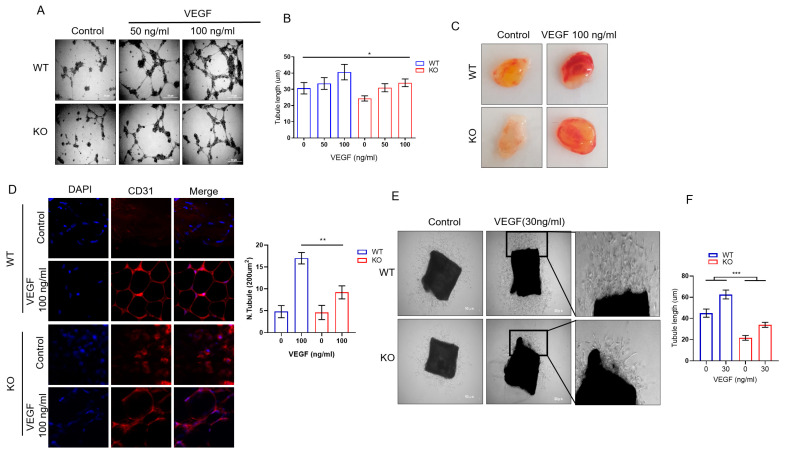
Effects of *DRG2* deficiency on the angiogenic function of endothelial cells. (**A**,**B**) Effects of *DRG2* deficiency on tube formation in mouse lung endothelial cells (mLECs). The mLECs from WT and *DRG2*^−/−^ (KO) mice were seeded on Matrigel and monitored for tube formation after incubation for 24 h in the absence or presence of recombinant VEGF-A (50 or 100 ng/mL). (**A**) Representative images of tube formation by mLEC cells stimulated with recombinant VEGF-A or the supernatant of melanoma cells. Scale bar = 50 μm. (**B**) Quantification of the length of tubes. Data represent the tube length from four randomly chosen fields of three independent experiments and mean ± SD. Two-way ANOVA, * *p* < 0.05. (**C**,**D**) Effect of *DRG2* deficiency on neovascularization in the Matrigel plug. Wild-type and *DRG2*^−/−^ (KO) mice were s.c. injected with Matrigel plugs containing 100 ng/mL VEGF, and blood vessels developed over 10 days. (**C**) Representative macroscopic images of Matrigel plugs from wild-type and *DRG2*^−/−^ mice. (**D**) Representative images of anti-CD31 stained paraffin sections of Matrigel plugs from wild-type and *DRG2*^−/−^ mice CD31 (original magnification, 400×) with quantification of CD31-positive vessels. A total of 3 fields from individual plugs of 3 animals per group was counted. The data are expressed as the mean ± SD (n = 9 in each group). ** *p* < 0.005. (**E**,**F**) Effect of *DRG2* deficiency on endothelial cell sprouting in an aorta ring assay. Aortic rings obtained from wild-type and *DRG2*^−/−^ mice were incubated for 7 days in the absence or presence of VEGF-A (30 ng/mL). (**E**) Representative areas of the aortic rings are marked with boxes and enlarged correspondingly on the right side of each panel (original magnification, 100×). (**F**) The graph represents the overall length of endothelial cell sprouts and the overall number of endothelial cell sprouts emerging from the aortic ring. The data are expressed as the mean ± SD (n = 6 aortic rings from two mice in each group). *** *p* < 0.001.

**Figure 3 ijms-23-02877-f003:**
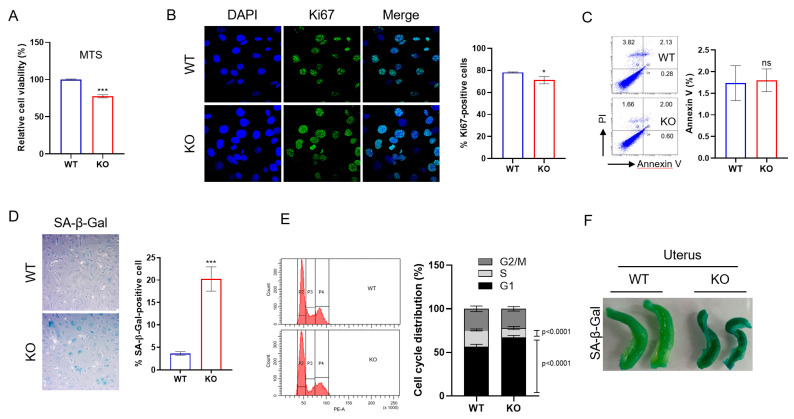
Effects of *DRG2* deficiency on the proliferation, apoptosis, and senescence of endothelial cells. Lung endothelial cells (mLECs) derived from wild-type and *DRG2*^−/−^ mice were cultivated under normal conditions as described in the Section 4 without any treatment. The mLECs were analyzed by (**A**) MTS assay (n = 3 independent experiments in each group) and (**B**) Ki67 staining for cell proliferation (n = 40 cells per 3 independent experiments in each group), (**C**) Annexin-V staining and FACS analysis for apoptosis (n = 3 independent experiments in each group), (**D**) senescence-associated β-gal (SA-β-gal) staining for cellular senescence (n = 120 cells per 3 independent experiments in each group), and (**E**) FACS analysis for cell cycle arrest (n = 10 independent experiments). (**F**) Detection of SA-β-gal activity in uteruses from wild-type and *DRG2*^−/−^ mice. Images in (**B**,**D**) are representative (Original magnification 600× and 200×, respectively) of Ki67-stained cells and SA-β-gal-stained cells, respectively. The data are expressed as the mean ± SD. * *p* < 0.05; *** *p* < 0.001. ns, not significant.

**Figure 4 ijms-23-02877-f004:**
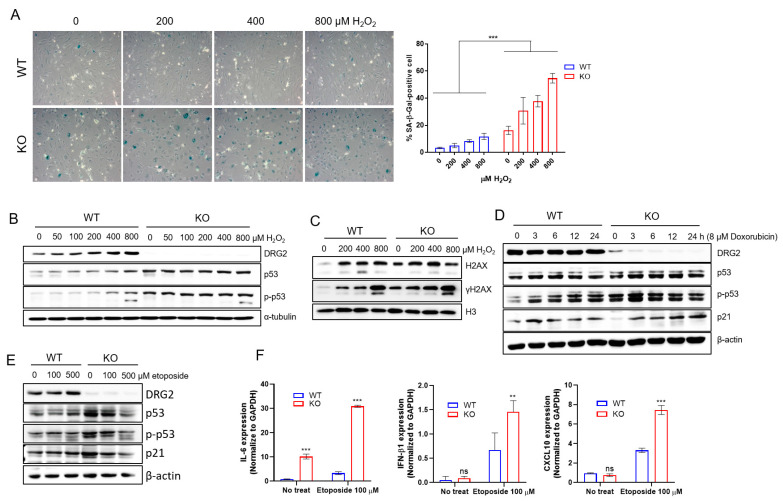
Enhanced oxidative stress-induced DNA damage and senescence in *DRG2* deficient endothelial cells. (**A**–**C**) mLECs from wild-type and *DRG2*^−/−^ mice were exposed to H_2_O_2_ at the indicated concentrations for 6 h. (**A**) Representative images of SA-β-gal-stained cells (original magnification 200×) with quantification of SA-β-gal-positive cells. The data are expressed as the mean ± SD (n = 5 in each group). *** *p* < 0.001. (**B**,**C**) Cells were analyzed by Western blot assay for the levels of (**B**) total and phosphorylated p53, and (**C**) γH2AX. (**D**) mLECs from wild-type and *DRG2*^−/−^ mice were treated with 8 μM doxorubicin for the indicated time and analyzed by Western blot assay for the levels of total and phosphorylated p53, and p21. (**E**) mLECs from wild-type and *DRG2*^−/−^ mice were treated with indicated concentration of etoposide for 24 h and analyzed by Western blot assay for the levels of total and phosphorylated p53, and p21. All Western blot data were representative ones of more than two experiments. (**F**) mLECs from wild-type and *DRG2*^−/−^ mice were incubated for 6 h in the absence or presence of 100 μM etoposide and analyzed by qPCR for levels of *IL-6*, *IFN-β1* and *CXCL10*. The data are expressed as the mean ± SD (n = 3 in each group). ** *p* < 0.005, *** *p* < 0.001.

**Figure 5 ijms-23-02877-f005:**
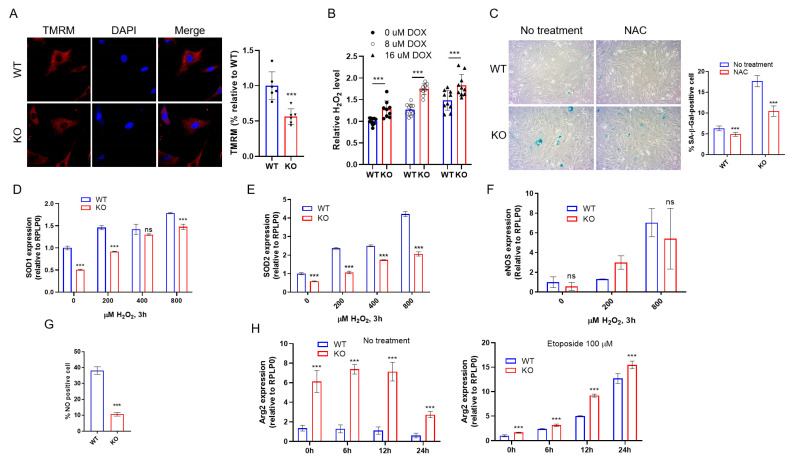
Effect of *DRG2* deficiency on the production of reactive oxygen species (ROS) and NO in endothelial cells. (**A**) representative images of wild-type and *DRG2*^−/−^ mLECs stained with TMRM for mitochondrial membrane potential and DAPI for nucleus (original magnification 600×). The data are expressed as the mean ± SD (n = 6 in each group). *** *p* < 0.001. (**B**) mLECs from wild-type and *DRG2*^−/−^ mice were exposed to doxorubicin at the indicated concentrations for 6 h and were analyzed for H_2_O_2_ levels as described in the Section 4. The level obtained from untreated wild-type mLECs was set to 1. The data are expressed as the mean ± SD (n = 10 in each group). *** *p* < 0.001. (**C**) Representative images of SA-β-gal-stained cells (original magnification 200×) with quantification of SA-β-gal-positive cells. mLECs from wild-type and *DRG2*^−/−^ mice were exposed to 5 mM NAC for 6 h and were analyzed for SA-β-gal-positive cells. The data are expressed as the mean ± SD (n = 3 in each group). *** *p* < 0.001. (**D**–**F**) Effect of *DRG2* deficiency on the H_2_O_2_-induced expression of anti-oxidant genes in endothelial cells. mLECs from wild-type and *DRG2*^−/−^ mice were exposed to H_2_O_2_ at the indicated concentrations for 6 h and analyzed by qPCR for the levels of *SOD1*, *SOD2*, and *eNOS*. The data are expressed as the mean ± SD (n = 3 in each group). ns, not significant. *** *p* < 0.001. (**G**) Effect of *DRG2* deficiency on the production of nitric oxide (NO) in endothelial cells. mLECs from wild-type and *DRG2*^−/−^ mice cultivated under normal condition without any treatment were analyzed for NO production as described in the Section 4. The data are expressed as the mean ± SD (n = 3 in each group). *** *p* < 0.001. (**H**) mLECs from wild-type and *DRG2*^−/−^ mice were incubated in the absence or presence of 100 μM etoposide for the indicated time and analyzed by qPCR for level of Arginase 2 (*Arg2*). The data are expressed as the mean ± SD (n = 3 in each group). *** *p* < 0.001.

## Data Availability

Not applicable.

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
