# Peer review of "DRG2 Depletion Promotes Endothelial Cell Senescence and Vascular Endothelial Dysfunction"

_ijms, 2022, doi:10.3390/ijms23052877_

Round 1
Reviewer 1 Report
The authors investigated the impact of DRG2 deficiency in endothelial cells. They showed that DRG2 KO triggers senescence of endothelial cells by increasing ROS production, and this associated with endothelial dysfunction in vivo, ex vivo and in vitro.
Minor typos:
- Cell analysis: « cells were fixed for “30 minutes overnight” in ice-cold 70% ethanol”. Is it 30 minutes or overnight?
- Line 491: a ROS not “an ROS”
- Line 354 : the 3 is missing (Fig 3C)
Results
- Fig 1A: in the legend it is indicated that 5 mice per group were analysed but there are only 3 mice on the dot plot for the wt
- Fig1B and C: How many mice were analysed? Is n the number of mice or the number of areas that were selected and counted?
- Fig 2D: where is the graph showing the quantification of CD31 positive cells as indicated in the legend?
- Fig 2E: what is the n in “n=6 in each group”?
- Fig3A: There is no mention of the MTS assay in material and methods
- Fig 3B: show the FACS data. Are n = 3 independent experiments?
- Fig 3C: Write annexin V instead of apoptosis on the y axis. Show the FACS data.
- Fig 3D: indicate how many cells were counted.
- DRG2 deficiency induces senescence. Measure the SASP in the DRG2-/- cells in comparison to the WT in absence of treatment with etoposide.
- Fig 4: Are the WBs shown representative of several independent experiments? Precise in the legend the n. If only one WB was done, it should be done at least a second time to confirm the results.
- Fig 5: The experiments don’t allow to conclude that the expression of Arg2 modulates the intracellular levels of NO and ROS. To prove this, the authors should do the experiment where they inhibit Arg2 in the DRG2-/- endothelial
Author Response
Comment 1: Cell analysis: « cells were fixed for “30 minutes overnight” in ice-cold 70% ethanol”. Is it 30 minutes or overnight?
Response 1: According to your comment, we corrected the sentence by removing “overnight” from the sentence.
Comment 2: Line 491: a ROS not “an ROS”
Response 2: According to your comment, we corrected the sentence by replacing “an” with “a” from the sentence.
Comment 3: Line 354: the 3 is missing (Fig 3C)
Response 3: According to your comment, we corrected the sentence by adding “3” to “(Fig. C)”.
Comment 4: Fig 1A: in the legend it is indicated that 5 mice per group were analysed but there are only 3 mice on the dot plot for the wt
Response 4: We analyzed 5 mice per each group. To make it clear we replaced the old graph with new one showing 5 dots for both WT and KO.
Comment 5: Fig1B and C: How many mice were analysed? Is n the number of mice or the number of areas that were selected and counted?
Response 5: The number “n” in Fig. 1B represents the number of mice and that in Fig. 1C represents number of counted areas of lungs from 2 mice in each group. According to your comment, we added this information to the Figure 1 legend.
Comment 6: Fig 2D: where is the graph showing the quantification of CD31 positive cells as indicated in the legend?
Response 6: According to your comment, we added the graph for the quantification of CD31 positive cells to the Fig. 2D.
Comment 7: Fig 2E: what is the n in “n=6 in each group”?
Response 7: The number “n” represents number of aortic rings from 2 mice in each group. According to your comment, we added this information to the Figure 1 legend.
Comment 8: Fig3A: There is no mention of the MTS assay in material and methods
Response 8: According to your comment, we added a subsection “2.11. MTS assay” in the “Materials and Methods” section. In this subsection, we described materials and methods used for the MTS assay.
Comment 9: Fig 3B: show the FACS data. Are n = 3 independent experiments?
Response 9: We made a mistake in describing the methods used for counting Ki67-positive cells. We actually used confocal microscopy but not FACS to quantify the Ki67-positive cells. We corrected the methods in “2.10. Ki-67 analysis” subsection by replacing “Fluorescence was analyzed using a FACSCanto II Flow Cytometer (BD Biosciences).” with “Confocal images were obtained using an Olympus 1000/1200 laser-scanning confocal system (Olympus, Shinjuku, Japan). All images were saved as TIFF files, and their contrast was adjusted with Image J (NIH, Bethesda, MD, USA, v1.19m). Quantification was performed by counting the number of Ki-67-positive cells.” “n=3” means 40 cells per 3 independent experiments. We modified the legend by providing this information.
Comment 10: Fig 3C: Write annexin V instead of apoptosis on the y axis. Show the FACS data.
Response 10: According to your comment, we replaced the Y axis-title “Apoptosis (%)” with “Annexin V (%)”. In addition, we provided the FACS data for the graph.
Comment 11: Fig 3D: indicate how many cells were counted.
Response 11: We counted total 360 cells (120 cells per 3 independent experiments). We modified the legend by providing this information.
Comment 12: DRG2 deficiency induces senescence. Measure the SASP in the DRG2-/- cells in comparison to the WT in absence of treatment with etoposide.
Response 12: We measured the SASP in WT and DRG2 -/- cells in the presence or absence of etoposide. According to your comment, we replaced the old “Fig. 4F” with new one containing SASP data from WT and DRG2 -/- cell in the presence or absence of etoposide. In addition, we provided additional description on this data in “Results” section.
Comment 13: Fig 4: Are the WBs shown representative of several independent experiments? Precise in the legend the n. If only one WB was done, it should be done at least a second time to confirm the results.
Response 13: We conducted the WB more than twice and the WB shown are representative of independent experiments. According to your comment, we indicated this information to the Figure 4 legend.
Comment 14: Fig 5: The experiments don’t allow to conclude that the expression of Arg2 modulates the intracellular levels of NO and ROS. To prove this, the authors should do the experiment where they inhibit Arg2 in the DRG2-/- endothelial.
Response 14: We think that your comment is critical and, thus, after receiving your comment, we tried to test the effect of Arg2 inhibition (by Arg2 inhibitor and siRNA against Areg2) on levels of NO and ROS in DRG2−/− mLECs. However, because of delay in delivering these materials, we could not conduct these experiment.
It has been widely accepted that, if L-arginine is depleted by increased level of Arg2, eNOS produces ROS instead of NO, which has been referred to as eNOS coupling. Even though we did not provide data for the effect of Arg2 inhibition on NO and ROS production in DRG2−/− mLECs, we provided several data supporting the increased occurrence of eNOS uncoupling in DRG2−/− mLECs: increased levels of Arg2 and ROS; decreased level of NO. These data, even though not perfect, may support our conclusion that the increased expression of Arg2 modulates levels of NO and ROS in DRG2−/− mLECs.
We add a brief description on this at “Discussion” section.
Reviewer 2 Report
In this study, the authors investigated the role of developmentally regulated GTP-binding protein 2 (DRG2) on endothelial cell senescence via DRG2 knockout (KO) mice. Several questions are raised and are suggested to be answered.
- The results demonstrated in Figure 1 are very surprised; the angiogenesis and micro-vessel density is significantly reduced in brain and lung tissue, I would like to know the micro-vessel density in liver and heart tissue in DRG2 knockout (KO) mice. Did the growth retardation or lethal effect observe in such significantly reduced angiogenesis mice?
- How about the senescence phenotype observed in DRG2 knockout (KO) mice?
- How about the reproduction in DRG2 knockout (KO) mice.
- The isolation of primary mLECs should be mentioned in the materials and methods.
- How to explain the DRG2 deficiency inhibits VEGF mediated angiogenic functions of endothelial cells?
- What is “Senescence-associated β-galactosidase activity assay” should be introduced.
- In line 358, these results indicate that DRG2 deficiency decreases proliferation and increases the senescence of endothelial cells. How to rule out the effect was not caused by cell cycle arrest? On the other hand, a cell growth simulator such as VEGF could be used in this part.
- In figure 4, the results demonstrated that the effect of DRG2 deficiency on oxidative stress-induced senescence of endothelial cells, did it mean that in DRG2 deficiency endothelial cells and anti-oxidant capability are reduced?
- In figure 5, the use of ROS scavenger N-acetylcysteine (NAC) to test whether increased levels of ROS lead to increased senescence in DRG2–/– mLECs. NAC treatment effectively reduced the number of SA-β-gal-stained cells in both wild-type and 421 DRG2–/– mLECs, did it mean that the DRG2 deficiency increased ROS in mLECs
- In figure 5, the results revealed that there was no significant difference in the expression of eNOS between DRG2–/– mLECs and wild-type mLECs. However, NO production was significantly decreased in DRG2–/– mLECs compared to wild-type mLECs. This is the most important point of the manuscript, the author mentioned that “Arg2 promoted eNOS uncoupling, decreased NO production, and increased ROS generation leading to endothelial dysfunction”, the statement should be described more clearly.
- In the final experiment, the result demonstrated that the expression level of Arg2 was significantly higher in DRG2–/– mLECs both before and after etoposide treatment than in wild-type mLECs, the vehicle alone group should be added.
- The relation between DRG2 and Arg2 could be emphasized in the topic; however, the role of DRG2 on endothelial cell senescence should be matched with the author mentioned “Cellular senescence is characterized by cell-cycle arrest and pro-inflammatory changes in gene expression”.
- What happen to mitochondria of DRG2–/– mLECs?
- While the endothelial cells response to stress environment, endothelial cells from DRG2 KO mice showed a senescence phenotype with decreased cell growth and enhanced levels of p21 and phosphorylated p53, γH2AX, dose it mean that in contrast, DRG2 may suppress levels of p21 and phosphorylated p53, γH2AX ?
- Overall, I think this is an important and valuable study to clarify the role of DRG2 in endothelial cells.
Author Response
Comment 1: The results demonstrated in Figure 1 are very surprised; the angiogenesis and micro-vessel density is significantly reduced in brain and lung tissue, I would like to know the micro-vessel density in liver and heart tissue in DRG2 knockout (KO) mice. Did the growth retardation or lethal effect observe in such significantly reduced angiogenesis mice?
Response 1: We did not determine the effect of DRG2 KO on the micro-vessel density in liver and heart. However, as you predicted, we think that DRG2 KO mice may have reduced micro-vessel density in other tissues such as liver and heart tissues.
Regarding the effect of DRG2 deficiency on the growth and lethalty, we previously reported the reduced growth and shortened lifespan of DRG2 KO mice (Int. J. Mol. Sci. 2020, 21, 60; doi:10.3390/ijms21010060). We added this information to the “Results” section.
Comment 2: How about the senescence phenotype observed in DRG2 knockout (KO) mice?
Response 2: According to your comment, we determined the SA-β-gal activity in uteruses from wild-type and DRG2–/– mice and found that SA-β-gal activity in uteruses from DRG2–/– mice was stronger than those from wild-type mice. We added data for SA-β-gal activity in uteruses as Fig. 3F). We also added description about the result.
Comment 3: How about the reproduction in DRG2 knockout (KO) mice.
Response 3: We found that the number of progeny per litter of DRG2 KO mice was significantly less than that of WT mice. According to your comment, we added the data for number of progeny per litter of WT and DRG2 KO mice (Fig. 1E) with description.
Comment 4: The isolation of primary mLECs should be mentioned in the materials and methods.
Response 4: According to your comment, we added description on the isolation of primary mLECs in the “Materials and Methods” section.
Comment 5: How to explain the DRG2 deficiency inhibits VEGF mediated angiogenic functions of endothelial cells?
Response 5: Proliferation of endothelial cells is one of the key steps of the VEGF-mediated angiogenic process (Unemori et al., 1992). DRG2 deficiency-induced senescence may limit the capacity of VEGF to promote angiogenesis in DRG2–/– endothelial cells. According to your comment, we added this information to “Discussion” section.
Comment 6: What is “Senescence-associated β-galactosidase activity assay” should be introduced.
Response 6: according to your comment, we modified the original sentences “We next asked whether DRG2 deficiency affects the senescence of endothelial cells. We evaluated senescence-associated β-gal (SA-β-gal) activity as a cellular marker of senescence in wild-type and DRG2–/– mLECs. “ to “We next asked whether DRG2 deficiency affects the senescence of endothelial cells. Senescent cells have increased activity of senescence-associated β-gal (SA-β-gal) (Dimri et al., 1995), a well-defined biomarker for cellular senescence. We evaluated SA-β-gal activity in wild-type and DRG2–/– mLECs.”
Comment 7: In line 358, these results indicate that DRG2 deficiency decreases proliferation and increases the senescence of endothelial cells. How to rule out the effect was not caused by cell cycle arrest? On the other hand, a cell growth simulator such as VEGF could be used in this part.
Response 7: We analyzed the cell cycle distribution of WT and DRG2 KO mLECs and found that DRG2 KO mLECs showed increased G1 distribution compared with WT mLECs. According to your comment, we added the cell cycle analysis data (Fig. 3E) with description as follow: One of the defining features of senescent cells is cell cycle arrest in G1 (Di Leonardo et al., 1994). We examined whether DRG2 deficiency induces cell cycle arrest in mLECs. Cell cycle distribution analysis revealed an increased accumulation of G1 phase cells in DRG2–/– mLECs (Fig. 3E).
Comment 8: In figure 4, the results demonstrated that the effect of DRG2 deficiency on oxidative stress-induced senescence of endothelial cells, did it mean that in DRG2 deficiency endothelial cells and anti-oxidant capability are reduced?
Response 8: Yes! To make it clear, we changed the title of Fig. 4 from “Effect of DRG2 deficiency on oxidative stress-induced senescence of endothelial cells.” to “Enhanced oxidative stress-induced DNA damage and senescence in DRG2 deficient endothelial cells.”
Comment 9: In figure 5, the use of ROS scavenger N-acetylcysteine (NAC) to test whether increased levels of ROS lead to increased senescence in DRG2–/– mLECs. NAC treatment effectively reduced the number of SA-β-gal-stained cells in both wild-type and 421 DRG2–/– mLECs, did it mean that the DRG2 deficiency increased ROS in mLECs
Response 9: Yes! To make it clear we modified the original sentence “increased ROS is one of the major causative factor for enhanced senescence in mLECs.” to “increased ROS is one of the major causative factor for enhanced senescence in DRG2–/– mLECs.”
Comment 10: In figure 5, the results revealed that there was no significant difference in the expression of eNOS between DRG2–/– mLECs and wild-type mLECs. However, NO production was significantly decreased in DRG2–/– mLECs compared to wild-type mLECs. This is the most important point of the manuscript, the author mentioned that “Arg2 promoted eNOS uncoupling, decreased NO production, and increased ROS generation leading to endothelial dysfunction”, the statement should be described more clearly.
Response 10: According to your comment, we described this sentence more clearly by modifying the original sentence “The increased ROS generation leads to endothelial dysfunction [47], [48].” to “eNOS produces NO by catalyzing L-arginine to L-citrulline. However, when the substrate L-arginine is depleted by the enzyme arginase (Arg), eNOS produces ROS instead of NO, which has been referred to as eNOS coupling. The increased ROS generation leads to endothelial dysfunction [47], [48].”
Comment 11: In the final experiment, the result demonstrated that the expression level of Arg2 was significantly higher in DRG2–/– mLECs both before and after etoposide treatment than in wild-type mLECs, the vehicle alone group should be added.
Response 11: according to your comment, we added a graph for the expression of Arg2 in wild-type and DRG2 KO mLECs without etoposide treatment (Fig. 5H).
Comment 12: The relation between DRG2 and Arg2 could be emphasized in the topic; however, the role of DRG2 on endothelial cell senescence should be matched with the author mentioned “Cellular senescence is characterized by cell-cycle arrest and pro-inflammatory changes in gene expression”.
Response 12: According to you comment, we added a data suggesting that DRG2 KO mLECs are arrested at G1 phase as Fig. 3E. We also added description about this result. We also modified two sentences as follows:
Original sentence “A characteristic feature of cellular senescence is the SASP [44] involving increased expression of inflammatory molecules [52].”
Modified sentence “A characteristic feature of cellular senescence is cell cycle arrest in G1 phase (Di Leonardo et al., 1994) and the SASP [44] involving increased expression of inflammatory molecules [52].”
Original sentence “Consistently, we observed increased levels of DNA damage marker γH2AX [43], p21 and phosphorylated p53 in DRG2–/– endothelial cells compared to wild-type endothelial cells.”
Modified sentence “Consistently, we observed cell cycle arrest in G1 phase (Di Leonardo et al., 1994) and increased levels of DNA damage marker γH2AX [43], p21 and phosphorylated p53 in DRG2–/– endothelial cells compared to wild-type endothelial cells.”
Comment 13: What happen to mitochondria of DRG2–/– mLECs?
Response 13: According to your comment, we added a data suggesting that DRG2 KO mLECs showed a decrease in mitochondrial membrane potential compared with wild-type mLECs (Fig. 5A). We also added description of the result.
Comment 14: While the endothelial cells response to stress environment, endothelial cells from DRG2 KO mice showed a senescence phenotype with decreased cell growth and enhanced levels of p21 and phosphorylated p53, γH2AX, dose it mean that in contrast, DRG2 may suppress levels of p21 and phosphorylated p53, γH2AX ?
Response 14: Even though we did not check the effect of DRG2 overexpression on the levels of p21 and phosphorylated p53, γH2AX, our results using DRG2 KO mLECs clearly suggest that DRG2 plays an important role to suppress senescence by decreasing the levels of p21 and phosphorylated p53, γH2AX in mLECs. We added a sentence “These results suggest that DRG2 deficiency increases the levels of p21, γH2AX, and phosphorylated p53 induced by oxidative stress in mLECs, which means that DRG2 may play a role in suppressing their levels in mLECs.” to result section.
Round 2
Reviewer 1 Report
I accept the revised version for publication.